**EMBO** *reports*

# Toll-like receptor 9 protects non-immune cells from stress by modulating mitochondrial ATP synthesis through the inhibition of SERCA2

Yasunori Shintani[1,2,*], Hannes CA Drexler[3], Hidetaka Kioka[4], Cesare MN Terracciano[5], Steven R Coppen[1], Hiromi Imamura[6], Masaharu Akao[7], Junichi Nakai[8], Ann P Wheeler[9], Shuichiro Higo[4], Hiroyuki Nakayama[10], Seiji Takashima[2,4], Kenta Yashiro[1] & Ken Suzuki[1,**]

## Abstract

Toll-like receptor 9 (TLR9) has a key role in the recognition of pathogen DNA in the context of infection and cellular DNA that is released from damaged cells. Pro-inflammatory TLR9 signalling pathways in immune cells have been well investigated, but we have recently discovered an alternative pathway in which TLR9 temporarily reduces energy substrates to induce cellular protection from stress in cardiomyocytes and neurons. However, the mechanism by which TLR9 stimulation reduces energy substrates remained unknown. Here, we identify the calcium-transporting ATPase, SERCA2 (also known as Atp2a2), as a key molecule for the alternative TLR9 signalling pathway. TLR9 stimulation reduces SERCA2 activity, modulating $Ca^{2+}$ handling between the SR/ER and mitochondria, which leads to a decrease in mitochondrial ATP levels and the activation of cellular protective machinery. These findings reveal how distinct innate responses can be elicited in immune and non-immune cells—including cardiomyocytes—using the same ligand-receptor system.

**Keywords** danger signal; DNA; SERCA2; TLR9
**Subject Categories** Metabolism; Immunology; Signal Transduction
**EMBO Reports (2014) 15, 438–445**

## Introduction

Extracellular DNA released from damaged tissue is a sign of danger. In the danger model [1,2], the self-molecules released from damaged cells (the damage-associated molecular patterns (DAMP)) are sensed by DAMP receptors on immune cells, triggering an inflammatory response. Toll-like receptors (TLRs) form a major group of DAMP receptors. Among TLRs, TLR9 is the only receptor for detecting DNA (self and non-self) [3], and hence, the DNA released from damaged cells can trigger inflammation via TLR9 in immune cells [2]. Naturally, TLR9 is expressed in immune cells; however, its expression was also reported in non-immune cells including cardiomyocytes and neurons [4,5]. It would be very damaging for such organs with poor regenerative capacity, if TLR9 in non-immune cells also operates the inflammatory signalling that increases tissue damage.

Recently, we have reported an alternative function of TLR9 to induce cellular protection in cardiomyocytes and neurons [6]. We found that CpG-oligodeoxynucleotide (CpG-ODN; TLR9 ligand) temporally reduces energy substrates to protect cardiomyocytes and neurons by activating AMP-activated protein kinase (AMPK) without inducing canonical inflammatory signalling. Comparison of the expression profiles between cardiomyocytes and macrophages and subsequent functional studies identified that the expression level of *Unc93b1* is a pivotal switch for the distinct TLR9 responses by regulating subcellular localization of TLR9. Unc93b1 is a chaperon-like protein that helps TLR9 travel from the ER to endosome to become the N-terminally shed, immune-prone type of receptor. After ligand stimulation, this cleaved TLR9 subsequently forms a signalling

1   William Harvey Research Institute, Barts and The London School of Medicine and Dentistry, Queen Mary University of London, London, UK
2   Department of Medical Biochemistry, Osaka University Graduate School of Medicine, Suita, Japan
3   Bioanalytical Mass Spectrometry, Max Planck Institute for Molecular Biomedicine, Muenster, Germany
4   Department of Cardiovascular Medicine, Osaka University Graduate School of Medicine, Suita, Japan
5   Laboratory of Myocardial Electrophysiology, Imperial Centre for Translational and Experimental Medicine, Imperial College London, National Heart & Lung Institute, Hammersmith Campus, London, UK
6   The Hakubi Center & Graduate School of Biostudies, Faculty of Medicine Campus, Kyoto University Science Frontier Laboratory building Room 305, Kyoto, Japan
7   Department of Cardiology, National Hospital Organization Kyoto Medical Center, Kyoto, Japan
8   Saitama University Brain Science Institute, Saitama City, Japan
9   Blizard Advanced Light Microscopy Facility, Blizard Institute, Barts and The London School of Medicine and Dentistry, Queen Mary University of London, London, UK
10  Laboratory of Clinical Science and Biomedicine, Osaka University Graduate School of Pharmaceutical Sciences, Osaka, Japan
    *Corresponding author. Tel: +81 668793492; Fax: +81 668793493; E-mail: yshintani@medbio.med.osaka-u.ac.jp
    **Corresponding author. Tel: +44 2078828236; Fax: +44 2078828256; E-mail: ken.suzuki@qmul.ac.uk

molecular complex with MyD88 to initiate inflammatory signalling in macrophages [7,8]. On the contrary, under low expression of *Unc93b1* in non-immune cells including cardiomyocytes and differentiated neurons, endocytosed DNA is transported to the ER via the retrograde route to bind the TLR9 that stays at the ER, subsequently decreases energy substrates and increases the AMP/ATP ratio, then activates AMPK [6]. However, the molecular mechanism by which TLR9 in the ER reduces intracellular ATP levels remains unknown.

## Results and Discussion

### SERCA2 is an adaptor for the alternative TLR9 signalling

The known inflammatory TLR9 signalling is mediated by a common TLR adaptor molecule, MyD88 [9]. However, we have recently demonstrated that the modulation of energy metabolism through TLR9 still operates in MyD88$^{-/-}$ cardiomyocytes [6], suggesting that this alternative TLR9 signalling is MyD88-independent and branches from the pro-inflammatory TLR9 signalling at the receptor level.

To identify adaptor molecules for the alternative, cellular protective TLR9 signalling, tandem affinity purification was performed in primary rat neonatal cardiomyocytes (alternative TLR9 signal; on) and cardiac fibroblasts for comparison (alternative TLR9 signal; off) using adenoviral vectors that encoded full-length TLR9 tagged with a human influenza hemagglutinin (HA)-FLAG at the C-terminus (TLR9-HA-FLAG) or Yellow fluorescent protein (YFP)-HA-FLAG. The comparison of TLR9 immunoprecipitates revealed the existence of a 95-kDa band associated with TLR9 in cardiomyocytes, but not in cardiac fibroblasts (Fig 1A). Importantly, intensity of this 95-kDa band was increased after CpG-ODN stimulation (Fig 1B). Mass spectrometric analysis identified this protein as sarcoplasmic reticulum (SR) Ca$^{2+}$ ATPase, SERCA2.

The association between TLR9 and SERCA2 was verified by a series of observations. First, reciprocal co-immunoprecipitation by SERCA2 demonstrated its binding to the overexpressed TLR9 in cardiomyocytes (Supplementary Fig S1A).

Second, to exclude the possibility that the SERCA2 and TLR9 association might be an artefact due to the TLR9 overexpression, we checked for endogenous association between TLR9 and SERCA2 using mouse neonatal cardiomyocytes treated with a cell-permeable crosslinker, dithiobis[succinimidylpropionate] (DSP) [10]. As shown in Fig 2A, TLR9 that was co-immunoprecipitated with SERCA2 was clearly detected in wild-type cardiomyocytes, but not in TLR9$^{-/-}$ cardiomyocytes, confirming that the association was unrelated to the overexpression of TLR9.

Third, to further confirm its specific binding, we added non-biased proteomics analysis of TLR9 immunoprecipitates from rat neonatal cardiomyocytes and cardiac fibroblasts. Most of heat shock proteins and ribosomal proteins were found in the immunoprecipitates from both cell types, which are major non-specific binding proteins from immunoprecipitates with an overexpressed bait (Supplementary Fig S1B) [11]. In this approach, we again confirmed SERCA2 to be a cardiomyocyte-specific TLR9-binding protein, while other abundant Ca$^{2+}$ pump proteins in cardiomyocytes, such as ryanodine receptor (RyR) or inositol 1,4,5-triphosphate receptor (IP$_3$R), were not detected in the immunoprecipitates from

cardiomyocytes (Supplementary Fig S1B). These data also support our finding that SERCA2 is a TLR9-associating protein.

Finally, immunofluorescence studies demonstrated that not only the overexpressed TLR9-HA-FLAG protein colocalized with SERCA2 and with the ER/SR marker KDEL in cardiomyocytes (Fig 2B and Supplementary Fig S1C), but also the endogenous TLR9 colocalized with SERCA2 (Fig 2C). Collectively from these data, we conclude that TLR9 interacts with SERCA2 at ER/SR in cardiomyocytes.

### TLR9 reduces SERCA2 activity and Ca$^{2+}$ content in the ER/SR in cardiomyocytes

To confirm the functional involvement of SERCA2 in the alternative TLR9 signalling, we performed a knockdown experiment of *SERCA2* in cardiomyocytes. Different siRNAs targeting *SERCA2* efficiently abolished the CpG-induced activation of AMPK (Fig 3A), suggesting that SERCA2 indeed plays a pivotal role in the alternative TLR9 signalling in cardiomyocytes.

SERCA2 is an ER/SR-resident Ca$^{2+}$-ATPase that governs Ca$^{2+}$ uptake from the cytosol to ER/SR and regulates Ca$^{2+}$ storage in the ER/SR. Given that SERCA2 has an enzymatic activity and that the binding between TLR9 and SERCA2 was enhanced by ligand stimulation (Fig 1B), we hypothesized whether TLR9 stimulation could modulate SERCA2 activity. First, by measuring Ca$^{2+}$-ATPase activity of SERCA2 in neonatal cardiomyocytes with or without CpG-ODN treatment, we found that TLR9 stimulation, indeed, significantly reduced SERCA2 activity (Fig 3B). Next, to directly monitor Ca$^{2+}$ in the SR/ER in living cardiomyocytes, we used a SR/ER-targeting Förster resonance energy transfer (FRET)-based Ca$^{2+}$ indicator, D1ER [12], by adenovirus-mediated delivery. Time-lapse observation demonstrated that CpG-ODN significantly reduced Ca$^{2+}$ in the SR/ER from 10 min after administration (Fig 3C and D). As Ca$^{2+}$ storage (Ca$^{2+}$ content) in the ER/SR controls the time course of the cytosolic Ca$^{2+}$ concentration during each cardiac beating cycle (Ca$^{2+}$ transient), next we analysed Ca$^{2+}$ content and Ca$^{2+}$ transient with or without CpG-ODN treatment using rat adult cardiomyocytes. SR Ca$^{2+}$ content measured by caffeine-induced amplitude of Ca$^{2+}$ transient (caffeine-induced F/F$_0$) was significantly reduced at 30 min after CpG administration (Fig 3E). As a consequence, the administration of CpG-ODN reduced the amplitude of Ca$^{2+}$ transient (F/F$_0$), Ca$^{2+}$ decay (mostly dependent on ER/SR Ca$^{2+}$ re-uptake) and time to peak (an indication of speed of ER/SR Ca$^{2+}$ release) in the Ca$^{2+}$ transient (Fig 3F and G). All these results indicate that SERCA2 activity is reduced by TLR9 stimulation.

Next, we tested whether SERCA2 inhibition is sufficient to switch on the alternative TLR9 signalling. To this end, we exploited the property of a low-molecular-weight compound, thapsigargin, as a SERCA2 inhibitor. We observed that short periods of thapsigargin treatment alone effectively increased AMPK phosphorylation in cardiomyocytes (Fig 3H), suggesting that modulation of Ca$^{2+}$ handling is sufficient to achieve the same outcome as that of the alternative TLR9 signalling. Thapsigargin is an irreversible SERCA2 inhibitor, and inhibiting SERCA2 by thapsigargin for hours is known to induce ER stress [13,14]. However, as demonstrated in Fig 3I, expression levels of *GRP78* and *GRP94* remained unchanged at 1 or 3 h after the administration of CpG-ODN, suggesting no substantial damage in the ER, presumably because CpG-ODN-induced decrease in SERCA2 activity was incomplete, transient and reversible [6].

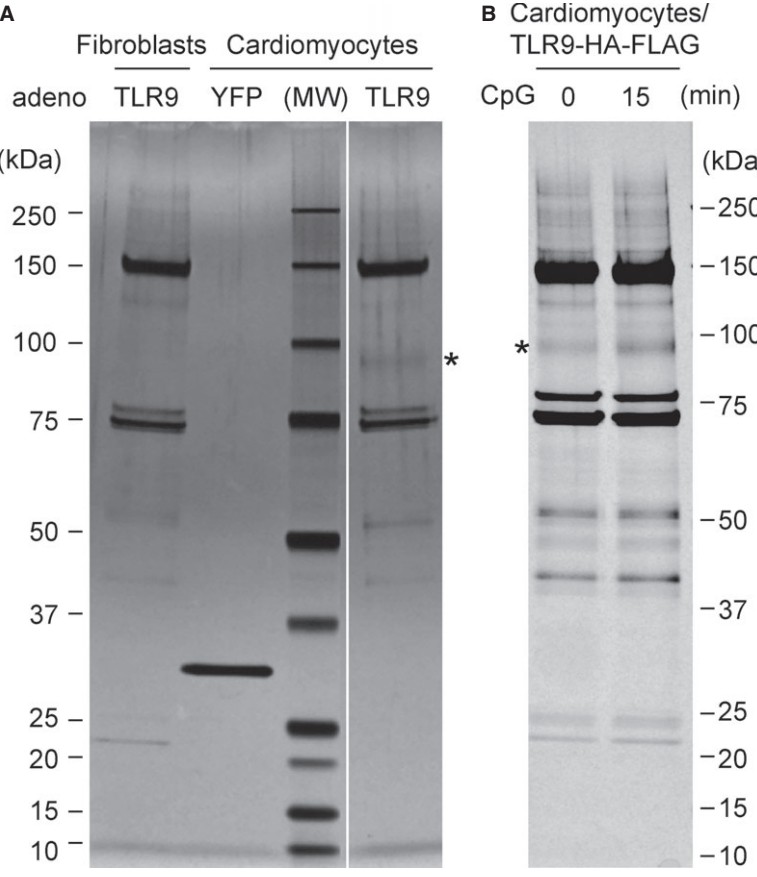

**Figure 1.  Identification of SERCA2 as a binding protein of toll-like receptor 9 (TLR9).**

A    Representative image of tandem affinity purification visualized by silver stain is presented. Tandem affinity purification was performed using cardiomyocytes that were adenovirally transfected with TLR9-HA-FLAG or YFP-HA-FLAG. Cardiac fibroblasts transfected with TLR9-HA-FLAG were used as a control. MW: molecular weight markers. Note that the 95-kDa band marked by an asterisk was observed in TLR9 immunoprecipitates from cardiomyocytes, but not from cardiac fibroblasts.

B    CpG-ODN increased the target protein binding to TLR9 (asterisk) in cardiomyocytes.

Source data are available online for this figure.

From these results, we conclude that SERCA2 is a functioning adaptor that mediates the alternative TLR9 signalling in cardiomyocytes.

**TLR9 reduces mitochondrial ATP synthesis by alteration of ER/mitochondria Ca$^{2+}$ handling via SERCA2**

The next question is how TLR9 reduces the ATP levels by decreasing SERCA2 activity. Not only several mitochondrial matrix dehydrogenases in the TCA cycle, but also almost all steps in oxidative phosphorylation (OXPHOS) are positively regulated by the level of mitochondrial Ca$^{2+}$ ([Ca$^{2+}$]$_m$) [15], and close proximity between ER/SR and mitochondria is a key for the regulation of [Ca$^{2+}$]$_m$ [16]. Immediately after Ca$^{2+}$ release from the ER/SR, local Ca$^{2+}$ concentration in the space between ER/SR and mitochondria will be high enough for the mitochondrial calcium uniporter to be activated. Therefore, Ca$^{2+}$ handling between the ER/SR and mitochondria plays a critical role in the regulation of mitochondrial ATP synthesis [17,18]. Hence, we examined whether TLR9 modulates Ca$^{2+}$ handling between these organelles in cardiomyocytes. First, using a

mitochondria-targeting Ca$^{2+}$ probe, G-CaMP2-mt [19], we found that [Ca$^{2+}$]$_m$ significantly dropped following the CpG-ODN administration (Fig 4A and B). In addition, thapsigargin administration without CpG-ODN effectively decreased [Ca$^{2+}$]$_m$ in cardiomyocytes (Supplementary Fig S2A), suggesting that the modulation of [Ca$^{2+}$]$_m$ in the alternative TLR9 signalling was mediated by SERCA2 inhibition.

Next, to further confirm a link between the decreased [Ca$^{2+}$]$_m$ and mitochondrial ATP synthesis, we measured mitochondrial ATP levels at the single-cell level using a mitochondria-targeting FRET-based ATP probe, mit-ATeam [20,21]. Adenovirus-mediated delivery of mit-ATeam enabled us to monitor the mitochondrial ATP levels (Fig 4C). Using this biosensor, we found that administration of CpG-ODN reduced the mitochondrial ATP levels in cardiomyocytes (Fig 4D), indicating that TLR9 can indeed decrease mitochondrial ATP synthesis. Collectively from these data, we concluded that TLR9 signalling in cardiomyocytes modulates energy metabolism by altering [Ca$^{2+}$]$_m$ via decreased SERCA2 activity, leading to the decrease in mitochondrial ATP synthesis, which subsequently switches on AMPK, whose activation leads to the increased stress tolerance [6].

    

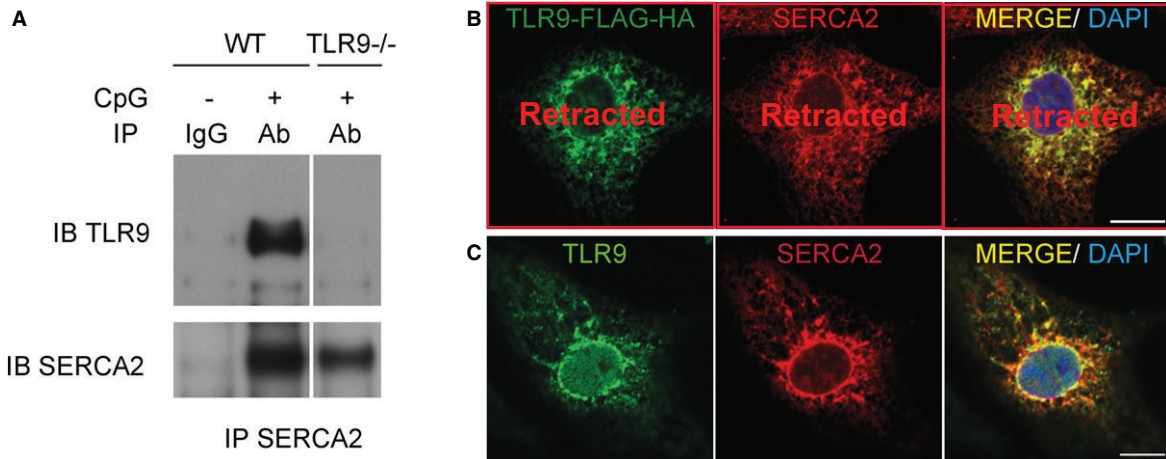

**Figure 2. SERCA2 is a functional adaptor for the alternative toll-like receptor 9 (TLR9) signalling in cardiomyocytes.**

A   Co-immunoprecipitated TLR9 with SERCA2 antibody after cross-linking with DSP was clearly detected in wild-type (WT) mouse neonatal cardiomyocytes. Co-immunoprecipitated TLR9 was abolished in cardiomyocytes from TLR9$^{-/-}$ mice, confirming the specificity of the detecting antibody. Each lane contains proteins immunoprecipitated from $5 \times 10^6$ cells.

B   ~~The overexpressed TLR9 colocalized with SERCA2 at the ER/SR. TLR9-HA-FLAG was expressed in cardiomyocytes and labelled with anti-HA antibody (green) in addition to SERCA2 (red) antibody.~~

C   Endogenous TLR9 colocalized with SERCA2 in cardiomyocytes. Rat neonatal cardiomyocytes were stained with TLR9 (green) and SERCA2 (red) antibodies at 15 min after CpG-ODN stimulation. The images were obtained by confocal microscopy. Scale bar indicates 10 μm. Experiments were repeated three times, and representative images are shown.

Source data are available online for this figure.

The main finding of this study is that SERCA2 is a key functioning adaptor/mediator for the alternative TLR9 signalling in cardiomyocytes. Our data showed that alternative TLR9 signalling decreased SERCA2 activity but did not cause detectable ER stress. As the alternative TLR9 signalling that decreases energy substrate and activates AMPK is transient [6], we speculated that SERCA2 inhibition by TLR9 is relatively short, reversible and partial, while in contrast thapsigargin is an irreversible SERCA2 inhibitor and causes ER stress when its inhibition is long-term. Indeed, it has been known that a number of drugs that reversibly inhibit mitochondrial respiration improve recovery from ischaemia-reperfusion injury [22].

While we used synthetic TLR9 ligand (CpG-ODN) throughout this study, mitochondrial DNA, which is rich in unmethylated CpGs and is therefore a potent ligand for TLR9, also induced AMPK activation in cardiomyocytes in the same manner as synthetic CpG-ODN [6]. Given that cardiomyocytes possess abundant mitochondria, it is likely that substantial amount of mitochondrial DNA would be released extracellularly in the heart upon tissue damage.

We previously demonstrated that differentiated neuronal cells also operate the alternative TLR9 signalling [6]. $Ca^{2+}$ uptake from the cytosol to ER is regulated by a ubiquitous/housekeeping splicing isoform of SERCA2, SERCA2b, in most cell types except for cardiomyocytes where SERCA2a is predominantly expressed [23]. By using the isoform-specific antibodies [24], we found that the both isoforms associated with TLR9 (Supplementary Fig S2B), suggesting that TLR9-SERCA2 association can be applied to other cell types. We have shown that the expression level of *Unc93b1* is a pivotal switch for the distinct TLR9 responses between cardiomyocytes and macrophage-like cell line RAW264.7; knockdown of

*Unc93b1* transformed the response to CpG-ODN in RAW264.7 cells from the inflammatory to the alternative TLR9 signalling, and *vice versa* [6]. Indeed, the interaction between TLR9 and SERCA2 became evident in *Unc93b1*-knocked-down RAW264.7 cells, while no such interaction in control cells (Supplementary Fig S2C). These data suggest that the stored $Ca^{2+}$ in the ER/SR is released from RyR in myocytes or from $IP_3R$ in other cell types upon stimulation and/or constitutive focal activation [17], and is then transferred to mitochondria through the local mitochondrial calcium uniporter. Both types of $Ca^{2+}$ release contribute to the maintenance of average $[Ca^{2+}]_m$ and hence mitochondrial bioenergetics.

In our previous report, TLR9 in cardiomyocytes was found only in the ER, but was not transported to the endosome regardless of CpG stimulation [6]. In addition, we were not able to detect cleaved TLR9 in cardiomyocytes. Further, the retrogradely transported DNA was found in the ER in the *Unc93b1*-knocked-down RAW264.7 cells, in which TLR9 cleavage was inhibited. Thus, we believe that retrograde transport of CpG-ODN does not require TLR9 cleavage and in cardiomyocytes TLR9 binds to CpG-ODN in the ER in an uncleaved form, subsequently interacting with SERCA2. According to the work of Miyake's group [25], a bona-fide inflammatory TLR9 receptor complex (TLR9N + TLR9C) requires *Unc93b1* expression and TLR9 cleavage in the endosome, which is consistent with our model. We also showed that TLR7 ligand also triggers the alternative TLR9 signalling as to AMPK activation, while TLR2 and four ligands do not induce the same response [6]. TLR3 and TLR7 also use Unc93b1-dependent transport system from the ER to endosome; therefore, it is likely that other intracellular TLRs are capable of interacting with SERCA2 in the cells with low expression level of *Ubc93b1*.

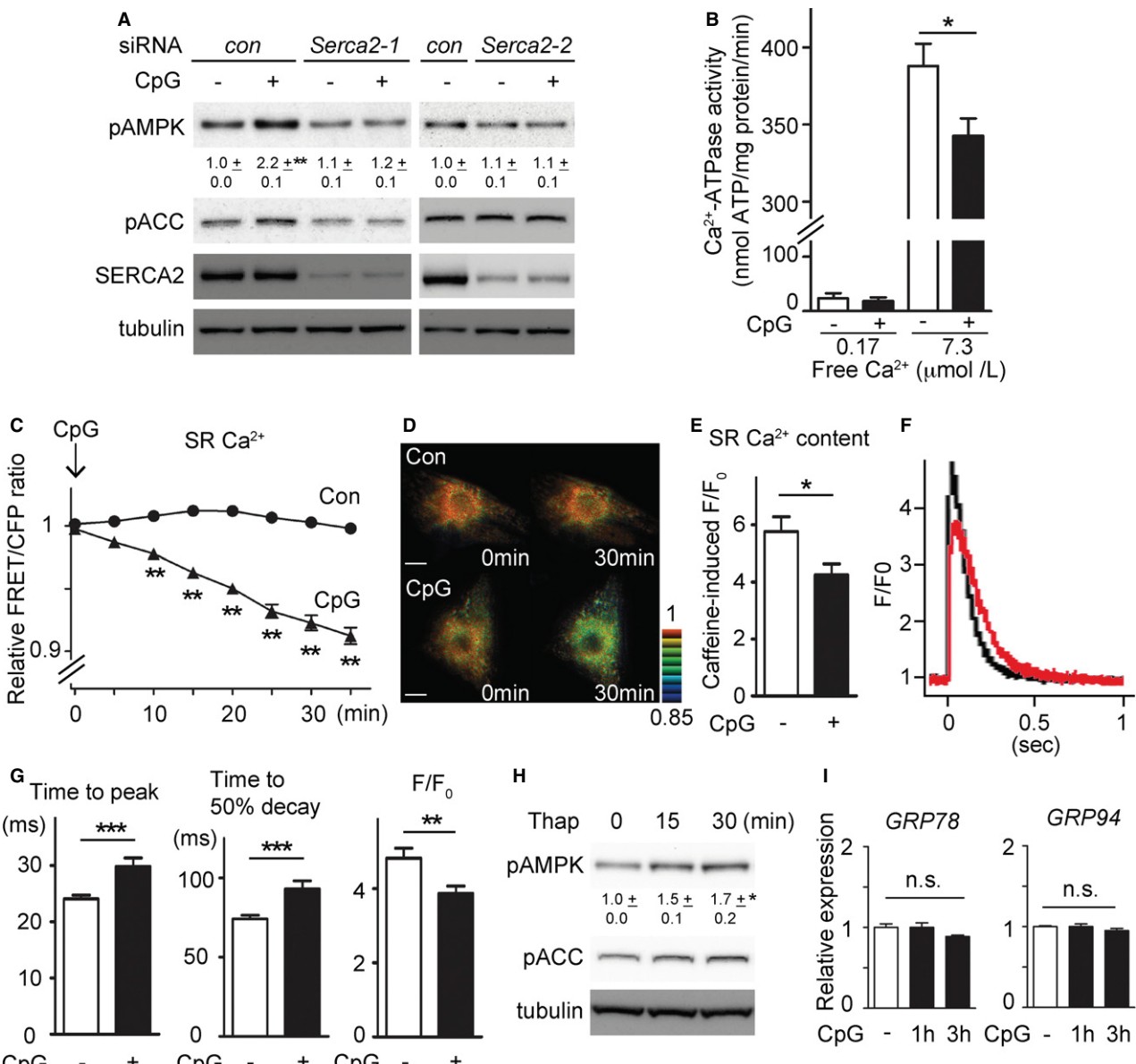

**Figure 3.  Toll-like receptor 9 (TLR9) stimulation decreases SERCA2 activity in cardiomyocytes.**

A   RNAi-mediated knockdown of *SERCA2* diminished the TLR9-induced AMP-activated protein kinase (AMPK) activation. Different siRNAs targeting SERCA2 similarly inhibited TLR9-induced AMPK activation. Values indicate densitometric ratio of pAMPK to tubulin in immunoblots, mean $\pm$ s.e.m. The data were obtained from three independent experiments.

B   $Ca^{2+}$-ATPase activity was assessed with microsome fractions from neonatal cardiomyocytes with or without 30-min treatment of CpG-ODN over two different concentrations of free $Ca^{2+}$. The data were obtained from 4 ($Ca^{2+}$; 0.17) or 5 ($Ca^{2+}$; 7.3) preparations in each group.

C   Förster resonance energy transfer (FRET)/CFP emission ratio plots of SR/ER $Ca^{2+}$ probe in cardiomyocytes treated with control (*n* = 26) or CpG-ODN (*n* = 27). CpG-ODN significantly reduced SR/ER $Ca^{2+}$ in neonatal cardiomyocytes after CpG-ODN. The measurements were normalized to the ratio at time 0.

D   Representative FRET/CFP ratiometric pseudocoloured images of SR/ER $Ca^{2+}$ probe in cardiomyocytes treated with control or CpG-ODN. Scale bar indicates 10 µm.

E   $Ca^{2+}$ content, as assessed by the amplitude of the caffeine-induced Fluo-4 transient in adult rat cardiomyocytes, was significantly decreased after CpG treatment (*n* = 28) compared to the control (*n* = 26).

F   Representative traces of cytosolic Fluo-4 transients from the control (black; *n* = 50) and CpG-ODN (red; *n* = 51)-treated cardiomyocytes elicited at 0.5 Hz pacing.

G   Time to peak and time to 50% decay of the Fluo-4 transients were both delayed by CpG stimulation. Also, the amplitude of the Fluo-4 transients (F/F$_0$) was decreased by CpG stimulation.

H   Thapsigargin (500 ng/ml for 0, 15, 30 min) increased pAMPK without the administration of CpG-ODN. The data were obtained from three independent experiments.

I   CpG-ODN administration did not induce ER stress that was assessed by the expression level of *GRP78* and *GRP94* and was measured by real-time PCR in rat cardiomyocytes treated with CpG-ODN for the indicated period.

Data information: $^{*}P$ < 0.05 $^{**}P$ < 0.01 $^{***}P$ < 0.001 compared with the control. Error bars indicate s.e.m.
Source data are available online for this figure.

**A**

Con

0min    30min

CpG

0min    30min

**B**

CpG

Mitochondrial Ca²⁺

Con

CpG

**

***

**C**

Mit-ATeam

**D**

Mitochondrial ATP

cFRET/CFP

Con

CpG

Oligo

**

***    **

***    ***

**E**

**Cardiomyocytes, Neurons**        **Inflammatory cells**

released DNA

endosome

**SERCA2**    **Retrograde transport**        endosome

Ca²⁺    **SERCA2 activity**    **TLR9**        **TLR9**    **MyD88**  **TRAF6**

ER/SR        **Unc93b1**

mito Ca²⁺ entry ↓    **Expression level of *Unc93b1***    **TLR9**

ATP synthesis ↓  Mito        ER/SR

**AMPK activation**        **Nucleus**

**Increased stress tolerance**        **Inflammatory response**

In this report, we uncovered the mechanism of the alternative TLR9 signalling in cardiomyocytes and we proposed a model that the same ligand-receptor system induces distinct biological responses upon tissue injury (Fig 4E). Mimicking the alternative TLR9 signalling, inhibition of mitochondrial ATP synthesis, might have therapeutic potential against myocardial infarction or other sterile inflammation. Notably, as the distinct TLR9 signalling pathways are separated at the receptor level, enhancing the protective effect will work synergistically with the inhibition of unnecessary canonical inflammatory signalling, for example MyD88 inhibition or *Unc93b1* knockdown.

◀

**Figure 4.  Toll-like receptor 9 (TLR9) decreases the mitochondrial ATP synthesis by altering [Ca²⁺]ₘ via SERCA2.**

A, B  Time-lapse analysis of mitochondrial matrix $Ca^{2+}$ in G-CaMP2-mt-transfected cardiomyocytes after the vehicle control ($n = 25$; circle) or CpG-ODN ($n = 20$; triangle) administration. CpG administration significantly reduced mitochondrial $Ca^{2+}$ in cardiomyocytes. Representative images with pseudocolour for each condition are shown in (A).

C  Adenovirus-mediated transfection of mit-ATeam showed mitochondrial localization in rat cardiomyocytes, which enabled to monitor the mitochondrial ATP levels.

D  Time-lapse analysis of mitochondrial ATP levels in mit-ATeam-transfected cardiomyocytes after the vehicle control ($n = 13$; circle), CpG-ODN ($n = 11$; triangle) or oligomycin ($n = 11$; square), an inhibitor of ATP synthase used as a positive control. Mitochondrial ATP levels significantly dropped after CpG stimulation in cardiomyocytes.

E  Schematic presentation of two different TLR9 signalling pathways. Cardiomyocytes or neurons sense DNA (danger signal) upon tissue injury, reduce energy metabolism by alteration of microdomain $Ca^{2+}$ handling through SERCA2 and consequently increase the stress tolerance through AMPK activation, while immune cells initiate the inflammatory response.

Data information: Scale bars indicate 10 μm. Error bars indicate s.e.m. **$P < 0.01$ ***$P < 0.001$ compared with the control.
Source data are available online for this figure.

## Materials and Methods

### Plasmids

Mouse TLR9-myc cDNA was a kind gift from Dr Brinkmann. C-terminal myc was replaced with HA-FLAG tag by a PCR-based method, and the sequence was confirmed.

### Tandem affinity purification

Cardiomyocytes or fibroblasts transfected with adenovirus expressing HA-FLAG-tagged TLR9 or YFP were lysed in lysis buffer (20 mM MOPS, pH 7.4, 10% glycerol, 0.15 M NaCl, 0.5% 3-[(3-Cholamidopropyl) dimethylammonio]-1-propanesulfonate (CHAPS), 1 mM EDTA, protease inhibitor cocktail (Sigma)) and immunoprecipitated with anti-FLAG M2 agarose (Sigma) at 4°C for 1 h. The beads were washed and eluted with buffer containing 0.25 mg/ml 3 × FLAG peptide (Sigma) at 4°C for 15 min. After removal of the FLAG-agarose, the eluates were immunoprecipitated with anti-HA matrix (Roche, clone 12CA5) at 4°C overnight. HA matrix was washed with lysis buffer followed by another wash with buffer containing 1 M NaCl and then eluted with 0.1 M glycine-HCl (pH 2.4) at room temperature for 10 min.

### Statistical analysis

The comparison between the two groups was made by *t*-test (two-tailed), and other comparisons were made by ANOVA followed by Bonferroni's *post hoc* test. Time-lapse data were assessed by two-way ANOVA repeated measure followed by Bonferroni's *post hoc*. A value of $P < 0.05$ was considered statistically significant. Additional methods can be found in the Supplementary Methods.

**Supplementary information** for this article is available online: http://embor.embopress.org

### Acknowledgements

We thank F. Wuytack for SERCA2a and b antibodies. We are grateful for the support to this project from the members of the Suzuki laboratory. The authors also thank for the technical support from Keiko Shingu and the members of the Takashima Lab on revision. This research was supported by the New Investigator Research Grant from Medical Research Council (G1000461 to Y.S.), Barts and the London Charity and NIHR-Cardiovascular Biomedical Research Unit (to K.S.).

### Author contributions

YS conceived and designed research; YS, HCAD and HK performed research; YS, HCAD, CMNT, SRC, HI, AWP, ST, KY and KS analysed data; HK, CMNT, ST, HI, MA, JN, HS, HN and AWP contributed reagents/analytic tools; YS and KS wrote the paper.

### Conflict of interest

The authors declare that they have no conflict of interest.

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
