## [Review Process File (PDF 876 KB) · EMBO Reports]

Manuscript EMBOR-2013-37945

TLR9 protects non-immune cells from stress by modulating mitochondrial ATP synthesis through the inhibition of SERCA2

Yasunori Shintani, Hannes C.A. Drexler, Hidetaka Kioka, Cesare M. N. Terracciano, Steven R. Coppen, Hiromi Imamura, Masaharu Akao, Junichi Nakai, Ann P. Wheeler, Shuichiro Higo, Hiroyuki Nakayama, Seiji Takashima, Kenta Yashiro and Ken Suzuki

Corresponding authors: Yasunori Shintani, Osaka University Graduate School of Medicine, Ken Suzuki, Queen Mary University of London

Review timeline:

Submission date:	04 September 2013
Editorial Decision:	01 October 2013
Revision received:	29 December 2013
Editorial Decision:	06 February 2014
Revision received:	07 February 2014
Accepted:	10 February 2014

Transaction Report:

Editor: Nonia Pariente

1st Editorial Decision

01 October 2013

Thank you for your submission to EMBO reports. We have now received reports from the three referees that were asked to evaluate your study, which can be found at the end of this email. As you will see, all the referees find the topic of interest and would be supportive of the publication of a suitably revised manuscript. In all, we think that if the referee concerns can be addressed, the study would be a good candidate for publication in EMBO reports.

Given that all referees provide constructive suggestions on how to strengthen the work, and provide evidence as to its physiological relevance and generality (whether other TLRs also mediate this pathway, as requested by referee 3), I would like to give you the opportunity to revise your manuscript. If the referee concerns can be adequately addressed, we would be happy to accept your manuscript for publication. However, please note that it is EMBO reports policy to undergo one round of revision only and thus, acceptance of your study will depend on the outcome of the next, final round of peer-review.

Revised manuscripts must be submitted within three months of a request for revision unless previously discussed with the editor; they will otherwise be treated as new submissions. Revised

manuscript length must be a maximum of 28,500 characters (including spaces). When submitting your revised manuscript, please also include editable TIFF or EPS-formatted figure files, a separate PDF file of any Supplementary information (in its final format) and a letter detailing your responses to the referees.

We also welcome the submission of cover suggestions or motifs that might be used by our Graphics Illustrator in designing a cover.

I look forward to seeing a revised form of your manuscript when it is ready. In the meantime, do not hesitate to get in touch with me if I can be of any assistance.

REFEREE REPORTS:

Referee #1 (Report):

This studies extends a recently published one from this group where it was discovered that toll-like receptor 9 (TLR9) activation by extracellular DNA triggered a stress-response promoting process in neurons and cardiomyocytes that involved TLR9 localization at the endoplasmic reticulum (ER), diminished intracellular ATP/AMP and enhanced AMPK activity. It was speculated that in these non-immune cells, an ER-mitochondrial communication triggered by TLR9 provided stress resistance through this AMPK activation mechanism.

Here they explored the mechanism. Using proteomics methods, they identified the ER/SR-Ca²⁺-ATPase SERCA2 as an interacting protein of ER-localized TLR9. This interaction was promoted by ligand activation of the receptor and was associated with diminished activity of the SERCA pump, fall of SR Ca²⁺ content and reduction of mitochondrial Ca²⁺ and ATP. A model is developed that extends one proposed by Cardenas et al, that constitutive Ca²⁺ transfer from ER to mitochondria, necessary for preserving mitochondrial bioenergetics, is disrupted by TLR9 interruption of this transfer by inhibition of the ability of the SERCA pump to maintain ER Ca²⁺ concentration.

1. According to this model, inhibition of SERCA by any means should increase AMPK phosphorylation. Indeed, the authors demonstrate that the SERCA inhibitor thapsigargin generates a similar phenotype. Unexpectedly, siRNA knockdown of SERCA2 did not similarly induce AMPK activation. How do the authors account for this result?
2. The overarching hypothesis is that SR Ca²⁺ is reduced by ligand activation of TLR9. However, SR [Ca²⁺] is not directly measured. Can the authors quantitatively estimate by how much the concentration is reduced?
3. The overarching hypothesis is that SR Ca²⁺ is reduced by ligand activation of TLR9 by inhibition of SERCA activity, but the activity is not directly measured, but needs to be.
4. Regarding the pacing experiments, it is unclear what transients are the focus of the statistical analyses. Shown is an example of the first transient of responses to 0.5 Hz repetitive stimulation. Are the statistics derived from the first transient for each cell for each pacing? Or from all transients during each pacing? It would be useful to demonstrate the average kinetic responses trace with error bars of all the responses.
5. The authors should explicitly point out that the effects of the TLR ligand on ATP and mito Ca²⁺ were observed in quiescent cells. Left unanswered in these studies is the relevance in actively-contracting cardiomyocytes, where Ca²⁺ dynamics are significantly modified. The lack of studies in actively contracting myocytes limits insights into the physiological relevance of the current studies.
6. Along similar lines, and also related to point 2 above, it would be quite useful to test the main hypothesis and conclusions by observing the kinetic responses of mito Ca²⁺ calcium concentration in response to release from stores before and after the TLR activation.

Referee #2 (Report):

In a very recent PNAS (USA) manuscript (MS) the authors described in cardiomyocytes and neuronal cells an unexpected role of TLR9 in energy metabolism and stress-protection. They

concluded that such non-immune cells (with low Unc93b1 expression) sense DNA, reduce energy consumption and thus increase stress tolerance (termed alternative TLR9 signalling, operating without MYD88, but including retrograde DNA transport to the ER).

Here the authors define molecular mechanisms by which TLR9 in the ER reduces intracellular ATP levels. Using state of the art tandem affinity purification, mass spectrometric analysis, immunofluorescence studies as well as non-biased proteomics, they identified in cardiomyocytes SERCA2 - endogenous as well when overexpressed - to interact with ligand activated TLR9. Knockdown of SERCA2 revealed that activated TLR9 reduced SERCA2 activity and thus caused an alteration of ER/mitochondrial Ca²⁺ handling. The authors conclude that SERCA2 functions as an adaptor for alternative TLR9 signalling. (both in neuronal cells and cardiomyocytes). This conclusion is novel and opens new ground.

While the techniques used and the results obtained are fine, the data raise questions:

1. Does the retrograde transport (from endosomes to ER) of the TLR9/Ligand complex require "cleaved" TLR9? Alternatively, uncleaved TLR9 could interact with SERCA2 within the ER. (see also the work of K. Miyake, *Nat. Commun.* 2013, 4:1949, and *International Immunology*, published February 26, 2013, not discussed).
2. Although this question refers to the PNAS MS, do the authors have quantitative expression data on Unc93b1 and MYD88 expression in cardiomyocytes versus immune cells (upon CpG exposure)?
3. The authors use thioated CpG ODN's as TLR9 ligand, their respective biology is at variance to that of natural DNA sequences.

Even though the data given are novel, and fine, the authors are urged to comment on the questions raised, since answers will impact on readers' understanding.

Referee #3 (Report):

Inflammation response triggered by Toll like receptor 9 (TLR9) signalling plays a positive and protective role in immune cells, but might impose quite negative and destructive effects on poorly regenerative tissues such as cardiomyocytes and neurons. Shintani's group identified an alternative TLR9 pathway safeguarding these cells by modulating energy metabolism. A pivotal switch for the distinct TLR9 response is Unc93b1, which controls trafficking of TLR9 from the endoplasmic reticulum (ER) to endolysosomes. In Unc93b1^{high} cells, TLR9 is transported to endosomes, where its association with ligand ODN-CpG will recruit MyD88 to activate inflammation. In Unc93b1^{low} cells, ODN-CpG translocates to the ER, binds to TLR9 there and reduces energy substrates. However, TLR9's adaptor in the alternative pathway remains unknown. In this manuscript, Shintani et al. demonstrated that in cardiomyocytes, the sarco/endoplasmic reticulum Ca²⁺-ATPase SERCA2 is the major adaptor molecule, and productively interacts with TLR9. The authors also showed that TLR9-SERCA2 association reduces SERCA2 activity, which consequently attenuates mitochondrial ATP synthesis. Overall, this work is interesting and extends our understanding of this field.

Major comments:

In Fig S2b, upon ODN-CpG stimulation, both SERCA2 isoforms associate with TLR9. The authors suggest "TLR9-SERCA2 association can be applied to other cell types."

Data from BIOGPS shows SERCA2 is a highly cardiocyte specific gene. By comparison its expression in immune cells, such as primary macrophages and macrophage-like cell line Raw264.7, is almost nothing. In their original paper on the alternative TLR9 pathway, the authors found Unc93b1 knockdown transformed the response to CpG-ODN in RAW264.7 cells from the inflammatory TLR9 signalling to the alternative TLR9 signalling that resulted in AMPK activation. How to explain this? What will play the role of SERCA2 in Raw264.7 cells?

Data on the expression and co-immunoprecipitation of TLR9 and SERCA2b in other cell types than the cardiomyocyte would be necessary to make such a suggestion.

Minor comments:

With the original PNAS paper and this work, the authors proposed a new TLR9 pathway in Unc93b1^{low} cells. TLR3 and TLR7 also utilize the Unc93b1 dependent transport system from ER to endosome. Like TLR9, they would be retained in the ER of cardiomyocytes. Are they also capable of interacting with SERCA2 in the Unc93b1^{low} cells? Does such an alternative pathway also exist downstream of these two TLRs?

Referee #1:

We appreciate Reviewer #1 for thoughtful and productive comments, which allowed us to improve the manuscript. We have addressed all the comments provided and believe that the manuscript is now much improved and solid.

1. According to this model, inhibition of SERCA by any means should increase AMPK phosphorylation. Indeed, the authors demonstrate that the SERCA inhibitor thapsigargin generates a similar phenotype. Unexpectedly, siRNA knockdown of SERCA2 did not similarly induce AMPK activation. How do the authors account for this result?

Thank you for the comment. We think that the results of these 2 experiments appeared to differ because of different time course after intervention. In the knockdown experiments TLR9 stimulation was conducted 48 hours after siRNA transfection, by which time the cells were allowed to adapt to the condition/environment. The basal mitochondrial $\text{Ca}^{2+}/\text{ATP}$ is compensated by other ion channels or transport from mitochondria, resulting in almost no difference in basal pAMPK. But, in these cells, TLR9 stimulation had no short-term effect on AMPK activation due to the lack of active SERCA2. On the contrary, we analysed the thapsigargin-treated cells 30-60 min after its administration, where the compensation mechanism had not completed yet, thereby leading to decrease in mito $\text{Ca}^{2+}/\text{ATP}$, and increase in pAMPK.

2. The overarching hypothesis is that SR Ca^{2+} is reduced by ligand activation of TLR9. However, SR $[\text{Ca}^{2+}]$ is not directly measured. Can the authors quantitatively estimate by how much the concentration is reduced?

Thank you very much for the productive comments. In the original manuscript, we measured SR Ca^{2+} contents by means of caffeine-induced Ca^{2+} transient (Fig. 3E in the revised manuscript), which is rather mobile fraction of SR/ER Ca^{2+} , and estimated the reduction by TLR9 stimulation to be 20%.

To further confirm our findings by another method, we directly monitored SR Ca^{2+} in spontaneously beating cardiomyocytes with time-lapse imaging by using SR-targeting Ca^{2+} probe (D1ER, [1]) according to the reviewer's suggestion. We confirmed significant decrease in the SR calcium at 30 min after TLR9 stimulation in cardiomyocytes. The results were added in the revised Fig 3C and D. We believe that these additional experimental data have strengthened our hypothesis.

3. The overarching hypothesis is that SR Ca^{2+} is reduced by ligand activation of TLR9 by inhibition of SERCA activity, but the activity is not directly measured, but needs to be.

Thank you for the constructive comment. We agree the reviewer's point. In this revised manuscript, we have additionally measured SERCA2 activity by assessing Ca^{2+} -ATPase activity as described previously [2]. The result was indeed supportive to our hypothesis that the membrane fraction from CpG-ODN treated cardiomyocytes showed significantly lower SERCA2 activity than that from control. We have added this result in Fig 3B.

4. Regarding the pacing experiments, it is unclear what transients are the focus of the statistical analyses. Shown is an example of the first transient of responses to 0.5 Hz repetitive stimulation. Are the statistics derived from the first transient for each cell for each pacing? Or from all transients during each pacing? It would be useful to demonstrate the average kinetic responses trace with error bars of all the responses.

We apologize for the confusion about the method we applied. Ca^{2+} transient experiment (Fig 3F and G in the revised manuscript) was not time-lapse experiment. We allowed 5 min

for cells to adapt to 0.5 Hz pacing and fluorescence across cells recorded by confocal microscopy using line scans. In off-line analysis, we selected 2 representative transients for each cell, which were almost identical in every beating, then measured parameters, which were averaged. We collected the data (control; $n=50$ and CpG-ODN; $n=51$ cells) for each condition from 3 independent preparations, and then subjected them into statistical analysis. We rewrote the method section more clearly in the supplementary information due to word limitation.

5. The authors should explicitly point out that the effects of the TLR ligand on ATP and mito- Ca^{2+} were observed in quiescent cells. Left unanswered in these studies is the relevance in actively contracting cardiomyocytes, where Ca^{2+} dynamics are significantly modified. The lack of studies in actively contracting myocytes limits insights into the physiological relevance of the current studies.

Thank you for the comments. Fig 3F and G in the revised manuscript was done in adult cardiomyocytes with 0.5 Hz stimulation, which means they were contracting, not quiescent. In addition, Mito- Ca^{2+} and ATP was measured in spontaneously-beating neonatal cardiomyocytes. The experiments in differentiated neurons were representation of static/quiescent cells. The points we would like to make is that regardless of the status of cells (beating or quiescent) the average mitochondrial calcium is the key determinant for mitochondrial ATP synthesis and it is affected by TLR9-SERCA2 axis.

6. Along similar lines, and also related to point 2 above, it would be quite useful to test the main hypothesis and conclusions by observing the kinetic responses of mito Ca^{2+} calcium concentration in response to release from stores before and after the TLR activation.

Thank you for the constructive comment. We have added new data showing SERCA2 activity and time-lapse monitoring of SR Ca^{2+} after TLR9 stimulation (section 2 and 3, above) in this revised manuscript, which, we believe, are convincing enough to support our hypothesis and we hope that the reviewer will agree. We agree that it would be interesting to add kinetic responses of mito Ca^{2+} , however, we don't have equipment and materials for conducting this experiment.

Referee #2:

We appreciate Reviewer #2 for her/his thoughtful and productive comments. We have addressed all the issues raised and believe that the manuscript is now much improved.

1. Does the retrograde transport (from endosomes to ER) of the TLR9/Ligand complex require "cleaved" TLR9? Alternatively uncleaved TLR9 could interact with SERCA2 within the ER. (see also the work of K. Miyake, Nat. Commun. 2013, 4:1949, and International Immunology, published February 26, 2013, not discussed).

Thank you very much for the comments and suggestion of the interesting works from Miyake's group. In our previous report, we demonstrated that TLR9 in cardiomyocytes was found only in the ER, but was not transported to the endosome regardless of CpG stimulation. In addition, we were not able to detect cleaved TLR9 in cardiomyocytes. Further, the retrogradely transported DNA was found in the ER in the *Unc93b1* knocked-down RAW264.7 cells, in which TLR9 cleavage was inhibited. Thus, we believe that retrograde transport of CpD-DNA does not require TLR9 cleavage and in cardiomyocytes TLR9 binds to CpG-ODN in the ER in an uncleaved form, subsequently interact with SERCA2. According to the work of Miyake's group, a bona-fide inflammatory TLR9 receptor complex (TLR9N + TLR9C) requires *Unc93b1* expression and TLR9 cleavage in the endosome, which is consistent with our model.

We added this discussion in the revised manuscript (page 7, paragraph 2).

2. Although this question refers to the PNAS MS, do the authors have quantitative expression data on *Unc93b1* and *MYD88* expression cardiomyocytes versus immune cells (upon CpG exposure)?

Thank you for the comments. Actually, we have shown these expression data in the previous manuscript. Please find the below figure.

Compared to the mild reduction in *MyD88* expression in cardiomyocytes, there was significant/striking difference in *Unc93b1* expression in cardiomyocytes compared to immune cells.

3. The authors use thioated CpG ODN's as TLR9 ligand, their respective biology is at variance to that of natural DNA sequences. Even though the data give are novel, and fine, the authors are urged to comment on the questions raised, since answers will impact on readers understanding.

Thank you very much for the comment and we agree the point. We showed in the previous manuscript that Type A CpG-ODN (phosphodiester) also activated AMPK in cardiomyocytes as similar to CpG-ODN type B (thioated), indicating that the AMPK activation is independent of CpG-ODN type. Also, to extend our findings with CpG-ODN, we demonstrated that mitochondrial DNA, which is rich in unmethylated CpGs and is therefore a potent ligand for TLR9, induced AMPK activation in cardiomyocytes in the same manner as synthetic CpG-ODN. Given that cardiomyocytes possess abundant mitochondria, it is likely that substantial amounts of mitochondrial DNA would be released extracellularly in the heart upon tissue damage.

We added this discussion in the supplementary discussion 1 due to word limitation.

Referee #3:

We appreciate Reviewer #3 for thoughtful and productive comments, which allowed us to improve the manuscript. We have addressed all the comments provided and believe that the manuscript is now much improved and solid.

Major comments:

In Fig S2b, upon ODN-CpG stimulation, both SERCA2 isoforms associate with TLR9. The authors suggest "TLR9-SERCA2 association can be applied to other cell types." Data from BIOGPS shows SERCA2 is a highly cardiomyocyte specific gene. By comparison its expression in immune cells, such as primary macrophages and macrophage-like cell line Raw264.7, is almost nothing. In their original paper on the alternative TLR9 pathway, the authors found *Unc93b1* knockdown transformed the response to CpG-ODN in RAW264.7 cells from the inflammatory TLR9 signalling to the alternative TLR9 signalling that resulted in AMPK activation. How to explain this? What will play the role of SERCA2 in Raw264.7 cells?

Data on the expression and co-immunoprecipitation of TLR9 and SERCA2b in other cell types than the cardiomyocyte would be necessary to make such a suggestion.

Thank you very much for the comment.

First, we can easily detect SERCA2 protein in RAW264.7 cells with Western blotting (Fig S2C). As the reviewer suggested, we have previously shown that the expression level of *Unc93b1* is a pivotal switch for the distinct TLR9 responses between cardiomyocytes and macrophage-like cell line RAW264.7; knockdown of *Unc93b1* transformed the response to CpG-ODN in RAW264.7 cells from the inflammatory to the alternative TLR9 signalling, and vice versa. Indeed, the interaction between TLR9 and SERCA2 became evident in *Unc93b1* knocked-down RAW264.7 cells, while no such interaction in control cells. We added this information in the revised manuscript (page 7, in red) and the figure in supplementary Fig S2C.

Regarding the expression level of *SERCA2* in other cell types, it seems that *SERCA2* expression (2a + 2b) is not cardiomyocytes-specific, rather ubiquitous and high copy number in cardiomyocytes. In cardiomyocytes, the contractile apparatus, which consists of actin-myosin fiber, always have the ER (for Ca^{2+}) and mitochondria (for ATP) in proximity, so that they can efficiently utilize Ca^{2+} and ATP for its continuous beating. Therefore, it is not surprising that *SERCA2* expression in cardiomyocytes will be far greater than that in other cell types. But this does not necessarily mean SERCA2 is not functioning in other cell types.

The difference between SERCA2a and 2b is based on the addition of 49 amino acids at the C terminus, and the affinity to Ca^{2+} of 2b is twice greater than that of 2a. It has been shown that calcium ions in the ER control cellular growth, proliferation, differentiation and death, and are maintained at high concentrations by the ubiquitous SERCA2b in almost all cells except for cardiomyocytes [4,5].

SERCA2 (sum of 2a and 2b) expression is indeed greater in heart than in other tissues according to the data from BIOGPS, as the reviewer pointed out. It is true that another openly available data set, Expression Atlas (EMBL-EBI, <http://www.ebi.ac.uk/gxa/home>), using RNA seq from different tissues also showed *SERCA2* expression is 20 times greater in heart than in leukocytes (556 vs 27).

Gene	adipose	adrenal	brain	breast	colon	heart	kidney	leukocyte	liver	lung	lymph node	ovary	prostate	skeletal muscle	testis	thyroid
ATP2A2	32	38	93	50	43	556	59	27	39	61	43	43	65	395	78	184

We checked another openly available dataset of mouse tissue exon array (GEO accession no. GSE21529, which was originally performed by BIOGPS team) [6]. They set almost each probe for one exon, suggesting more comprehensive coverage of gene expression. According to this data, although *SERCA2* was again found to be expressed more in the heart compared other tissues, *myosin heavy chain 6 (myh6)*, well-known cardiomyocytes-specific gene, demonstrated more significant/clear heart-specific expression pattern, which is quite different from that of *SERCA2* (please see below).

This was also confirmed by Expression Atlas data (see below).

Gene	adipose	adrenal	brain	breast	colon	heart	kidney	leukocyte	liver	lung	lymph node	ovary	prostate	skeletal muscle	testis	thyroid
MYH6						204								16	0.8	4

These information suggest that *SERCA2* expression (2a + 2b) is not cardiomyocytes-specific, rather ubiquitous and high copy number in cardiomyocytes.

Minor comments:

With the original PNAS paper and this work, the authors proposed a new TLR9 pathway in *Unc93b1*low cells. TLR3 and TLR7 also utilize the *Unc93b1* dependent transport system from ER to endosome. Like TLR9, they would be retained in the ER of cardiomyocytes. Are they also capable of interacting with SERCA2 in the *Unc93b1*low cells? Does such an alternative pathway also exist downstream of these two TLRs?

We agree with the point the reviewer made. In the previous PNAS paper, we have shown TLR7 ligand also triggers the alternative TLR9 signalling as to AMPK activation, while TLR2 and 4 ligands do not induce the same response (Please see Fig, below). As the reviewer pointed out, TLR 3 and 7 also use *Unc93b1*-dependant transport system from the ER to endosome, therefore it is likely that other intracellular TLRs are capable of interacting with SERCA2 in the cells with low expression level of *Ubc93b1*. We added this discussion in the supplementary discussion 2 due to word limitation.

Fig. TLR7 (Imiquimod) clearly showed AMPK activation at 60 min after the administration, while there was no AMPK activation after TLR2 and 4 stimulation [3].

References.

1. Palmer AE, Jin C, Reed JC, Tsien RY (2004) Bcl-2-mediated alterations in endoplasmic reticulum Ca²⁺ analyzed with an improved genetically encoded fluorescent sensor. *Proc Natl Acad Sci USA* **101**: 17404–17409.
2. Münch G (2002) Evidence for Calcineurin-mediated Regulation of SERCA 2a Activity in Human Myocardium. *J Mol Cell Cardiol* **34**: 321–334.
3. Shintani Y, Kapoor A, Kaneko M, Smolenski RT, D'Acquisto F, Coppen SR, Harada-Shoji N, Lee HJ, Thiemeermann C, Takashima S, et al. (2013) TLR9 mediates cellular protection by modulating energy metabolism in cardiomyocytes and neurons. *Proc Natl Acad Sci USA* **110**: 5109–5114.
4. Berridge MJ, Bootman MD, Roderick HL (2003) Calcium: Calcium signalling: dynamics, homeostasis and remodelling. *Nat Rev Mol Cell Biol* **4**: 517–529.
5. Vandecaetsbeek I, Trekels M, De Maeyer M, Ceulemans H, Lescrinier E, Raeymaekers L, Wuytack F, Vangheluwe P (2009) Structural basis for the high Ca²⁺ affinity of the ubiquitous SERCA2b Ca²⁺ pump. *Proc Natl Acad Sci USA* **106**: 18533–18538.
6. He A, Kong SW, Ma Q, Pu WT (2011) Co-occupancy by multiple cardiac transcription factors identifies transcriptional enhancers active in heart. *Proc Natl Acad Sci USA* **108**: 5632–5637.

2nd Editorial Decision

06 February 2014

Very many thanks for your patience while your revised study has been under peer-review. It was sent back to reviewers 1 and 2, and reviewer 1 has just sent in his/her report. As you will see in their remarks pasted below, they both now strongly support publication of your study in EMBO reports. I am therefore writing with an 'accept in principle' decision, which means that I will be happy to

accept your manuscript for publication once a few minor issues/corrections have been addressed, as follows.

- Please note that basic Materials and Methods required for understanding the experiments performed (including the whole Statistical analysis section) must remain in the main text, although additional detailed information may be included as Supplementary Material.

- In addition, we cannot accommodate supplementary discussion. Please ensure that the crucial points currently present in these discussions (such as the possibility that this pathway applies to other TLRs) are included in the main text.

- Please ensure that all figure legends include information on what the bar represents (mean, median), the type of error bars used, statistical test applied to the data and the level of significance indicated by the asterisks used. If the information applies to more than one panel in a figure, please include it at the end of the figure legend and specify the panels for which it is applicable.

- In order to accommodate the above requested inclusions, we can increase our length to a maximum of 30,000 characters, including spaces.

- In a routine check, we have realized that the two paragraphs of the introduction and the second half of the third paragraph on page 6 are very similar to text from your recent PNAS study on the non-canonical stress-protective role of TLR9. Although I appreciate you are also the author of that study, in order to prevent potentially embarrassing problems after publication, I must ask you to please rewrite these sections so they are not so similar.

- Lastly, we now encourage the publication of original source data -particularly for electrophoretic gels and blots, but also for graphs- with the aim of making primary data more accessible and transparent to the reader. If you agree, you would need to provide one PDF file per figure that contains the original, uncropped and unprocessed scans of all or key gels used in the figures and an Excel sheet or similar with the data behind the graphs. The files should be labelled with the appropriate figure/panel number, and the gels should have molecular weight markers; further annotation could be useful but is not essential. The source files will be published online with the article as supplementary "Source Data" files and should be uploaded when you submit your final version. If you have any questions regarding this please contact me.

After all remaining corrections have been attended to, you will receive an official decision letter from the journal accepting your manuscript for publication in the next available issue of EMBO reports. This letter will also include details of the further steps you need to take for the prompt inclusion of your manuscript in our next available issue.

Thank you for your contribution to EMBO reports.

REFeree REPORTS:

Referee #1:

The authors have nicely addressed all the comments I raised in my initial review.

Referee #2:

The amended version of this MS has incorporated - as much as I can see - satisfying answers to questions raised by the three referees. Given their previous report (PNAS, 2013) describing a novel "alternative Pathway" in which TLR9 temporarily reduces energy substrates in Cardiomyocytes and Neurons, here the authors identify the calcium transporting ATPase SERCA2 as TLR9 binding adaptor molecule, that modulates CA "handling" between SR/ER and mitochondria thus reducing mitochondrial ATP levels, and thus causing cellular protection. The data are novel, well

executed, and very interesting. Since the same receptor-Ligand system is now shown to induce distinct biological responses - canonical proinflammatory responses versus inhibition of mitochondrial ATP synthesis - mimicking of the novel "alternative pathway" may have therapeutic potential against myocardial infarction.

2nd Revision - authors' response

07 February 2014

Response to the editor's comment

- Please note that basic Materials and Methods required for understanding the experiments performed (including the whole Statistical analysis section) must remain in the main text, although additional detailed information may be included as Supplementary Material.

We did it according to the instruction.

- In addition, we cannot accommodate supplementary discussion. Please ensure that the crucial points currently present in these discussions (such as the possibility that this pathway applies to other TLRs) are included in the main text.

We moved them in the main text that were marked in red.

- Please ensure that all figure legends include information on what the bar represents (mean, median), the type of error bars used, statistical test applied to the data and the level of significance indicated by the asterisks used. If the information applies to more than one panel in a figure, please include it at the end of the figure legend and specify the panels for which it is applicable.

Yes, we believe all legends are ok.

- In order to accommodate the above requested inclusions, we can increase our length to a maximum of 30,000 characters, including spaces.

Final character count was 29991.

- In a routine check, we have realized that the two paragraphs of the introduction and the second half of the third paragraph on page 6 are very similar to text from your recent PNAS study on the non-canonical stress-protective role of TLR9. Although I appreciate you are also the author of that study, in order to prevent potentially embarrassing problems after publication, I must ask you to please rewrite these sections so they are not so similar.

Thank you for the comment. Page 6 last part was deleted so that we could integrate additional discussion which was originally put in the supplementary discussion. Page 3 first paragraph, we changed it as different as possible. These are shown in green in revised version.

- Lastly, we now encourage the publication of original source data -particularly for electrophoretic gels and blots, but also for graphs- with the aim of making primary data more accessible and transparent to the reader. If you agree, you would need to provide one PDF file per figure that contains the original, uncropped and unprocessed scans of all or key gels used in the figures and an Excel sheet or similar with the data behind the graphs. The files should be labeled with the appropriate figure/panel number, and the gels should have molecular weight markers; further annotation could be useful but is not essential. The source files will be published online with the article as supplementary "Source Data" files and should be uploaded when you submit your final version. If you have any questions regarding this please contact me.

We uploaded these pdfs and 1 excel file which contains the data of Figures 3B_3C_3E_3G_4B_4D. This time, we realized Fig 1A is actually a cropped and merged image from the same gel, which was now shown in Source Data file for Figure 1. We added a white line in the main Figure 1A. I sincerely apologized that I missed this point when I initially submitted it.

3rd Editorial Decision

10 February 2014

I am very pleased to accept your manuscript for publication in the next available issue of EMBO reports. Thank you for your contribution to our journal.

As part of the EMBO publication's Transparent Editorial Process, EMBO reports publishes online a Review Process File to accompany accepted manuscripts. As you are aware, this File will be published in conjunction with your paper and will include the referee reports, your point-by-point response and all pertinent correspondence relating to the manuscript.

If you do NOT want this File to be published, please inform the editorial office within 2 days, if you have not done so already, otherwise the File will be published by default [contact: emboreports@embo.org]. If you do opt out, the Review Process File link will point to the following statement: "No Review Process File is available with this article, as the authors have chosen not to make the review process public in this case."

Thank you again for your contribution to EMBO reports and congratulations on a successful publication.